# The Mu3e experiment

**F. Wauters*** **on behalf of the Mu3e collaboration**

PRISMA+ Cluster of Excellence and Institute of Nuclear Physics,
Johannes Gutenberg Universität Mainz, Germany

⋆ fwauters@uni-mainz.de

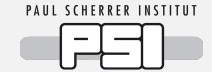 *Review of Particle Physics at PSI*

## Abstract

**The *Mu3e* experiment aims for a single event sensitivity of $2 \cdot 10^{-15}$ on the charged lepton flavour violating $\mu^+ \to e^+ e^+ e^-$ decay. The experimental apparatus, a light-weight tracker based on custom High-Voltage Monolithic Active Pixel Sensors placed in a 1 T magnetic field is currently under construction at the Paul Scherrer Institute, where it will fully use the intense $10^8\ \mu^+$/s beam available. A final sensitivity of $1 \cdot 10^{-16}$ is envisioned for a phase II experiment, driving the development of a new high-intensity continuous muon source which will deliver $>10^9\ \mu^+$/s to the experiment.**

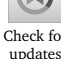
## 20.1 Introduction

Searches for Charged Lepton Flavour Violation (CLFV) in muon decays are a remarkably sensitive method to search for new physics processes [1]. These decays are free from Standard Model backgrounds, and leave a relatively simple and clear signature in the experimental apparatus. In addition, intense muon beams are available at several facilities, where the relatively long-lived muons get transported from a production target to an experimental area.

The Paul Scherrer Institute (PSI) has been at the forefront of CLFV searches, with the current best limit on the $\mu^+ \to e^+\gamma$ decay channel of $4.2 \cdot 10^{-13}$ (90% CL) from the *MEG* experiment [2]. The *SINDRUM* experiment [3] set the best limit on the $\mu^+ \to e^+ e^+ e^-$ decay channel, and the *SINDRUM II* experiment [4] on muon conversion $\mu^- \to e^-$ on gold. A new generation of experiments pursuing these three *golden* channels, which probe for new physics in a complementary manner [5], is currently under construction: the *Mu2e* experiment at Fermilab, the *COMET* experiment at J-PARC, and the *MEGII* experiment at PSI. The *Mu3e* experiment aims for a $10^{-16}$ single-event sensitivity for the $\mu^+ \to e^+ e^+ e^-$ CLFV decay channel, an improvement by four orders of magnitude compared to the limit set by the SINDRUM experiment [3]. Such a leap in sensitivity is enabled by the availibility of high-intensity muon beams, the use of silicon pixel detectors instead of multi-wire proportional chambers to track the decay products, and a modern data-aqcuisition system able to handle the vast amount of

data producted by the detector at high beam rates. A first phase of the experiment is currently under construction at the $\pi$E5 beamline at PSI, where the intense DC surface muon beam of $10^8 \ \mu^+/s$ will be exploited to achieve a single event sensitivity of $2 \cdot 10^{-15}$ in 300 days of data taking [6].

The *Mu3e* detector is optimized for the $\mu^+ \to e^+ e^+ e^-$ decay. It is designed to track the two positrons and one electron from muons decaying at rest with a light-weight tracker placed inside a 1 T magnetic field, thereby reconstructing the decay vertex and invariant mass. The momentum balance of the three reconstructed particles should be consistent with a muon decaying at rest. Several background processes can potentially meet the same criteria as the reconstructed signal events. The dominating accidental background originates from the overlay of two ordinary muon decays where one of the positrons produces an additional electron track through Bhabha scattering in the target material. This process is sufficiently suppressed by means of a good vertex resolution of better than 300 $\mu$m, a timing resolution of a few 100 ps, the requirement of an invariant mass equal to the muon mass, and a balanced momentum budget. Additional background from $\mu^+ \to e^+ e^+ e^- \nu_e \bar{\nu}_\mu$ internal conversion decays can only be suppressed by means of an excellent momentum resolution of $\sigma_p < 1$ MeV , as shown in Figure 20.1.

All *Mu3e* detector sub-systems, as described in Section 20.2, are currently under construction. With the solenoid magnet (Figure 20.2) installed at PSI, the first engineering runs are planned for spring 2021.

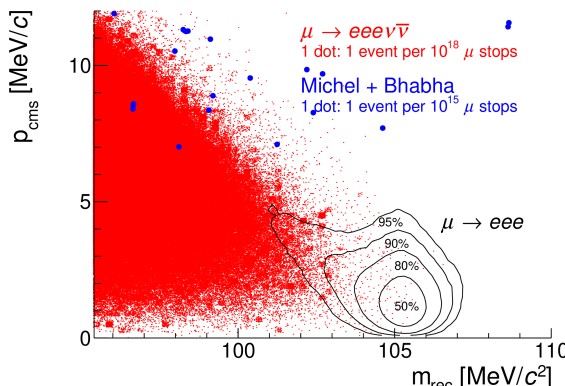

Figure 20.1: The simulated reconstructed mass versus the momentum balance of two positrons and one electron from a common vertex [6]. The accidental background is shown in blue, the dominating background from internal conversion is shown in red.

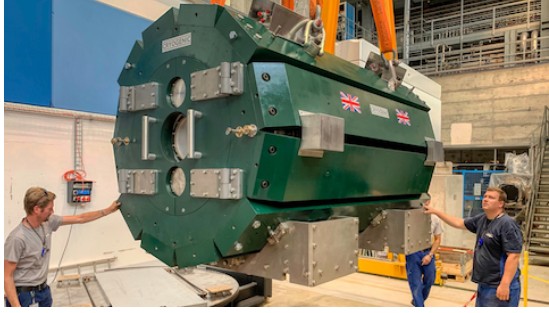

Figure 20.2: The 30 ton *Mu3e* magnet arriving at PSI. The magnet is curently installed and commissioned in the $\pi$E5 experimental area, providing a magnetic field of up to 2.6 Tesla with a $\frac{\Delta B}{B}$ uniformity and stability of $\mathcal{O}(10^{-4})$.

## 20.2 The Mu3e detector

The *Mu3e* detector is located at the Compact Muon Beam Line at the $\pi$E5 channel. After the positron contamination is removed from the beam by a Wien filter, the surface $\mu^+$ beam of up to $10^8 \ \mu^+/s$ is transported to the center of the *Mu3e* solenoid magnet, and stopped on a hollow double-cone target, which spreads out the decay vertices in $z$ and minimises the amount of target material traversed by the decay particles. The target is surrounded by the cylindrical central tracker, consisting of the inner silicon pixel detector, a scintillating fibre tracker for time measurements, and the outer silicon pixel detector. A momentum resolution of better than 1 MeV/c is achieved by letting the positrons(electrons) recurl in the magnetic field,

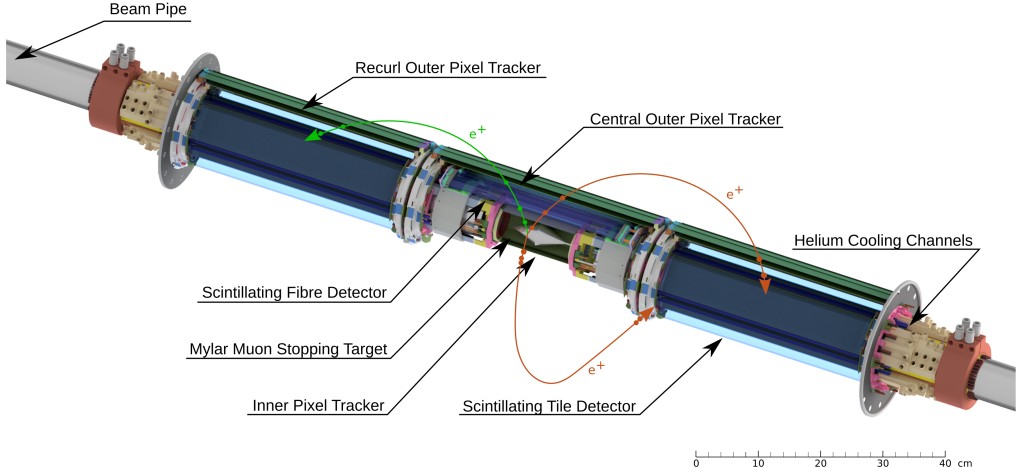

Figure 20.3: The active part of the *Mu3e* detector, with a central tracker surrounding the target, and upstream and downstream outer pixel tracking stations. The large lever arm created by the recurling tracks enables the high momentum resolution required.

either crossing the central tracker again, or hitting the outer tracking stations surrounding the upstream and downstream beam pipe. These stations consist of a silicon pixel tracker, and a scintillating tile detector mounted on the inside of the pixel tracker. The 5 mm thick tiles enable a time resolution for the tracks reaching these outer stations of better than 100 ps. The active part of the *Mu3e* detector is depicted in Figure 20.3.

As multiple Coulomb scattering is the dominating factor affecting the momentum resolution, it is crucial to minimize the material budget in the tracking detectors. For this purpose, the collaboration has developed a custom High-Voltage Monolithic Active Pixel Sensor [7] (HV-MAPS) based on a commercial 180 nm HV-CMOS process. After a series of prototypes, the sensor showed good efficiency (>99%) and time resolution ($\mathcal{O}(10$ ns)) [8] [9]. The *Mu3e MuPix* HV-MAPS is a 2x2 cm$^2$ sensor with 80x80 $\mu$m$^2$ active pixels, thinned to 50 $\mu$m (Figure 20.4). The digital periphery provides up to three 1.25 Gbit/s Low-Voltage Differential Signaling (LVDS) continuous data connections to the front-end electronics. The sensors are bonded to a thin aluminum/polyimide flex print carrying all electrical signals. Together with a polyimide support structure, the entire silicon tracking module has a thickness of ca. 0.0012 radiation lengths. The pixel sensors generate about 250 mW/cm$^2$ of heat. To remove this heat whilst keeping the material budget of the tracker sufficiently low, a gaseous He cooling system [10] is deployed providing well controlled He flows at atmospheric pressure between and outside the pixel layers.

A time resolution of about 10 ns is insufficient to determine the direction and thus the charge of the decay particles. A scintillating fibre detector is therefore placed between the inner and outer layer of the central silicon-pixel tracker, consisting of a dozen 30 cm long ribbons made from three staggered layers of 250 $\mu$m diameter multiclad round fibers, read out by Silicon Photomultipliers (SiPM) arrays on both sides [11]. Located at the very end of the recurling particle trajectories hitting the upstream or downstream tracker, where the constraints on the material budget are less stringent, the tile detector provides the needed precise timing information of the particle tracks, in conjunction with the fibre detector significantly reducing the accidental background associated with the intense rate of 10$^8$ $\mu^+$/s. Each of the 5824 individually wrapped tiles is read out by a single SiPM. Both the fibre and tile SIPM signals are processed by a custom Application-Specific Integrated Circuit (ASIC), the 32 channel *MuTrig*

chip [12], which applies 2 thresholds to the analogue signal for time and energy information. The *MuTrig* chip has a 1.25 Gbit/s LVDS data connection, similar to the *MuPix* chip readout. For tile and fibre detector a respective time resolution of <50 ps and <400 ps is achieved.

The entire *Mu3e* detector is mounted in the bore of a superconducting magnet. Figure 20.2 shows the 3 m long solenoid magnet with the iron return yoke. It has a 1 m wide bore housing the active detector, in addition to the support structures and services such as the front-end readout electronics and DC-DC power converters for the detector ASICs. The two flanges below and above the beam pipe provide access for the water and gaseous helium cooling pipes, the power cables, and the optical data connections.

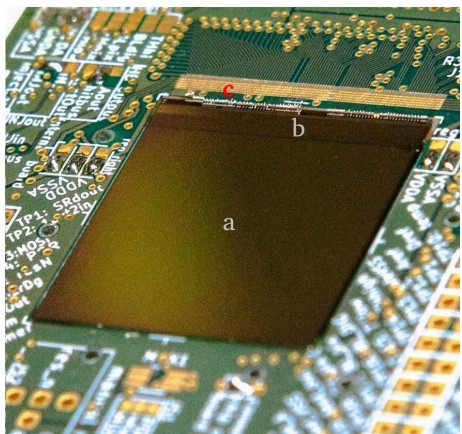

Figure 20.4: The full sized MuPix sensor, with a) a 2x2 cm$^2$ sized active area, and b) a periphery with the pixel hit digitization and readout state machine. This chip is c) wire bonded to a PCB for testing purposes.

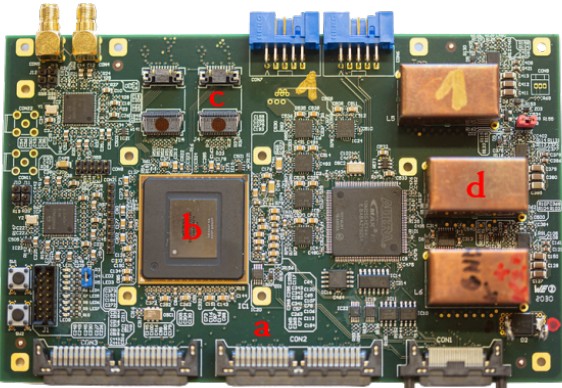

Figure 20.5: The front-end readout board, combining and time sorting the a) data from up to 36 detector ASICs on b) an Arria V FPGA, before sending the data via the c) optical Samtec FireFly tranceivers. d) Custom DC-DC converters with air coils regulate the power on the board.

## 20.3   Readout and online event selection

With three lepton tracks going in different (opposite) directions, the topology of a $\mu^+ \rightarrow e^+e^+e^-$ event is such, that a global picture of the detector is needed before candidate events can be selected. This leads to a trigger-less readout scheme as shown in Figure 20.6, where all pixel, fibre and tile hits are continuously being digitized and merged into a data stream of up to 100 Gbit/s. A series of PC's housing powerful Graphics Processing Units (GPU) perform an online event-selection, reducing the data rate to a manageable 50-100 MByte/s which is stored for further offline processing.

Each detector ASIC, a *MuTrig* or *MuPix* chip, assigns a timestamp and address to each hit, and sends the serialized data through a series of flex-prints and twisted pair cables to a front-end board (Figure 20.5). Each of these readout boards is located inside the magnet bore and accepts up to 45 electric LVDS links. The data streams are merged and time-sorted on an Arria V Field-Programmable Gate Array (FPGA). Two optical transceivers provide eight 6 GBit/s links to the outside, sending off the merged and sorted hit information combined with the slow-control data. In addition, the front-end FPGA also configures the detector ASICs, including tuning the very large number of individual *MuPix* pixels, and distributes the clock and reset signals.

All incoming and outgoing data connections to and from the detector volume travel via optical fibres to the counting house. The data links from the 112 front-end boards are con-

nected to the *Switching boards*, where the data from different detector modules are merged into 64 ns time slices containing the full detector hit information. This custom *PCIe40* board housing a large Arria 10 FPGA and 48 fast optical receivers and 48 fast optical transmitters was developed for the LHCb and ALICE upgrades [13].

The online event selection must decide which of these 64 ns *snapshots* of the detector to store for later (offline) processing, in the process keeping less than 1% of the data. A simple time coincidence between 3 tracks is insufficient to achieve this. Instead an online filter farm reconstructs all tracks in software, and performs the selection by requiring 3 tracks having a common vertex and the kinematics of a possible $\mu^+ \rightarrow e^+e^+e^-$ event. The filter farm consists of 12 PC's housing a FPGA board receiving the data and a powerful commercial GPU performing the event selection. With simple geometric cuts, candidate tracks are first selected on the FPGA from hits in the central pixel tracker. The track fitting [14] is performed on the GPU, where $1 \cdot 10^9$ fits per second have been achieved on a NVIDIA GTX 980 GPU, sufficient to be able to process the expected $10^8$ muon decays/s. A newer more powerful GPU will be selected when equipping the farm PCs.

The MIDAS[1]-based data-acquisition system sends the filtered data to on-site and off-site storage for later processing. This integrated DAQ also takes care of the configuration, monitoring, and logging of all parameters of the detector and its services such as the water and helium cooling system and power distribution.

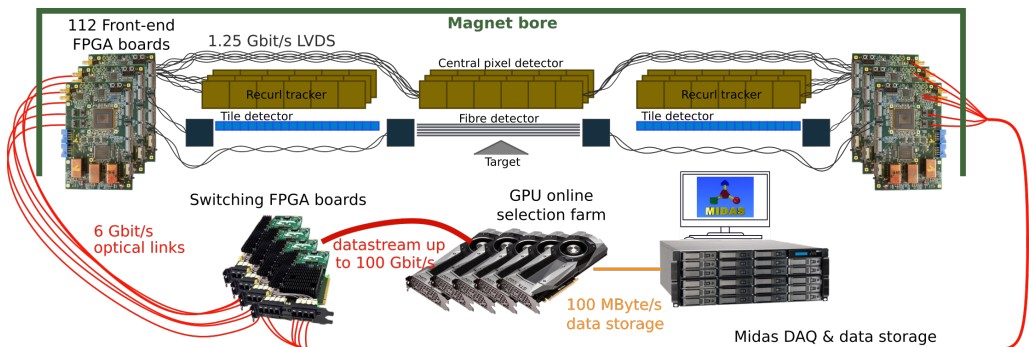

Figure 20.6: A sketch of the *Mu3e* triggerless readout scheme [15], where all detector hits are piped to the online filter farm. A selection algorithm based on massive parallelised track fitting sends off a subset of the data for further offline processing.

## 20.4 Conclusions and outlook

With the magnet installed at the Paul Scherrer Institute, the Mu3e experiment is entering its construction phase. All sub-detector demonstrators have met the required specification, and are currently being integrated to a single lightweight electron/positron tracker. This also includes a novel read-out system of the apparatus, which pipes the full detector information to an online filter farm. Aside from being a necessary requirement set by the CLFV decay event topology, this readout scheme where the full and global detector information is available for online analysis, also allows other new-physics searches such as CLFV two-body decays and Dark Photon searches [16].

The *Mu3e* phase II experiment envisions a branching ratio sensitivity of $1 \cdot 10^{-16}$. Many detector sub-systems are already designed with this goal in mind, but significant research and development on the detector side still has to be done. An order of magnitude increase in

---

[1]https://midas.triumf.ca

sensitivity also requires a more intense, and currently unavailable muon flux of $\mu^+$/s of $\mathcal{O}(10^9)$. For this purpose, a new High-Intensity Muon Beamline [17] to be installed at the target M is currently under development at the Paul Scherrer Institute, replacing the conventional muon extraction beamline elements with solenoids. The timeline of this project coincides with the envisioned start of the *Mu3e* Phase II construction at the end of this decade.

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
