# Peer review of "The Mu3e experiment"

_SciPost Physics Proceedings, doi:SciPost Phys. Proc. 5, 020 (2021)_

## Round 1 · Referee Report · Adrian Signer · 2021-2-26

Report

We (the editors Cy Hoffman, Klaus Kirch, Adrian Signer) had the
opportunity to review an earlier draft of the article and were in
communication with the authors before the submission. All our comments
and suggestions have been taken into account. Hence, we think the
paper can now be published in the current form.

---

## Round 1 · Referee Report · Anonymous · 2021-4-9

Report

The article is clear, sometime a bit too dense. To fit even better into the planned volume, the article could put more weight on the reasoning of the design concept of the Mu3e experiment (e.g. what was the
limiting factor in the SINDRUM measurement; what are the challenges of the measurement and what are the main features of the detector concept). At the
same time some aspects in the detector description (e.g. the history of the MuPIX chip) are probably less important for a reader to get the conceptual ideas.
Beside a few minor corrections I recommend the article for publication.

Requested changes

Figures: If not generated for this article, the figures should be referenced properly in the figure caption.

Detailed comments:

-----to the Abstract:

l 9: for / on ? (the charged lepton ...)

l 14: phase II experiment "at a muon beam with an intensity of 10^9 muons / sec",

------to the Introduction

l 26: I suggest to end the sentence with a ":" after construction -> "construction: the Mu2e experiment at Fermilab, .... "

l 28: aims for "an ultimate" single-event sensitivity of 10^-16 for the ...

l 33 - l 41: You should mention that main aim of the detector is to reconstruct the invariant muon mass and to measure the momentum of the decaying
muon with outstanding resolution and how the signal events are identified. You can be a bit more explicit here. Currently the paragraph is very dense!

l 35: "accidental" - you have not talked about backgrounds, thus "accidental" cannot be understood here.

l 35: originates from "the overlay of" two...

l 36: an additional electron "track"

l 36/37: through Bhabha scattering "on electrons of the target material"

l 39: The background from -> "Additional background from radiative decays with subsequent photon conversion"

l 40 : drop "internal conversion"

l 41: You should discuss the Figure as it is "the heart" of the experimental concept. "Higher resolution measurement of the momentum of the decaying muon together with the reconstruction of the invariant mass allows the separation between signal and backgrounds"

Figure 20.3/ caption: "upstream and downstream tracking stations" are notions not used in the legend of the figure.

----- to Mu3e detector

l 52: "timing purpose" -> time measurement

l 53: "is achieved" -> will be achieved

l 62-65: rephrase and drop the reference to MuPIX10
(it will anyhow be MuPIX11 in the final detector)

l 68: "0.0012 radiation length" -> 0.12 % of a radiation length

l 73: of about 10ns "as provided by the pixel sensor"

l 73/74: "to determine the the direction and thus the charge of the decay particle" -> meaning is not clear - you should explain better what you mean here.

l 77: Acronym "SiPM" not introduced

l 80: notion "combinatorial background" not introduced (only accidental and converted photons)

l 84 "energy" -> pulse height

---- to Readout and online

l 98: "powerful Graphic" (space missing)

l 116: snapshots -> better "tine slices" (introduced above)

l 124: "were achieved" -> have been achieved

-----to the Conclusion:

l 132: May add: "In spring 2021", with the magnet installed......

l 140: "branching ratio" sensitivity of "for the decay \mu->eee"

l 142: "Such an order" -> "An order"

  • validity: -
  • significance: -
  • originality: -
  • clarity: -
  • formatting: -
  • grammar: -

Author:  Frederik Wauters  on 2021-04-28  [id 1388]

(in reply to Report 2 on 2021-04-09)

We thank the referee for the constructive comments, suggestions, and corrections. The manuscript was adapted accordingly.

Will be resubmitted shortly.

Some particulars: * We acknowledge that the article somewhat dense * A sentence was added to clarify/contrast the Mu3e experiments and the SINDRUM experiment * A few sentences were added to the experimental concept, single-versus-background section to increase readability * The referee several times mentions " (measurement of the) momentum of the decaying muon". We measure muon decay at rest. This was stressed again to further clarify. * The developments of the HV-MAPS and the streaming DAQ are crucial to make Mu3e possible. I added a reference to a DAQ overview paper we published in the mean time. * "it will anyhow be MuPIX11 in the final detector" --> correct, but we are currently installing a v1 vertex tracker with MuPiX 10 chips. As the sensor shown in figure 20.4 is besides a few changes/correction, equal to the 'final' sensor, I dropped the prototype numbering from the manuscript. The reader can go to the references for further details. * "energy" -> pulse height . No, we measure TOT, not pulse height. I prefer to keep "energy", as this is agnostic to the technical implementation.

---

## Editorial Decision

published